# Effects of the Combination of β-Hydroxy-β-Methyl Butyrate and R(+) Lipoic Acid in a Cellular Model of Sarcopenia

**DOI:** 10.3390/molecules25092117

**Published:** 2020-04-30

**Authors:** Lorenzo Di Cesare Mannelli, Laura Micheli, Elena Lucarini, Carmen Parisio, Alessandra Toti, Barbara Tenci, Matteo Zanardelli, Jacopo Junio Valerio Branca, Alessandra Pacini, Carla Ghelardini

**Affiliations:** 1Department of Neurosciences, Psychology, Drug Research and Child Health-Neurofarba-Pharmacology and Toxicology Section, University of Florence. Viale Pieraccini 6, 50139 Florence, Italy; laura.micheli@unifi.it (L.M.); elena.lucarini@unifi.it (E.L.); carmen.parisio@unifi.it (C.P.); alessandra.toti@unifi.it (A.T.); barbara.tenci@unifi.it (B.T.); matteo.zanardelli@unifi.it (M.Z.); carla.ghelardini@unifi.it (C.G.); 2Department of Experimental and Clinical Medicine, Anatomy Section, University of Florence, Largo Brambilla 3, 50134 Florence, Italy; jacopojuniovalerio.branca@unifi.it (J.J.V.B.); alessandra.pacini@unifi.it (A.P.)

**Keywords:** thioctic acid, muscle wasting, myotube, oxidative stress, leucine

## Abstract

Sarcopenia is a clinical problem associated with several pathological and non-pathological conditions. The aim of the present research is the evaluation of the pharmacological profile of the leucine metabolite β-hydroxy-β-methyl butyrate (HMB) associated with the natural R(+) stereoisomer of lipoic acid (R(+)LA) in a cellular model of muscle wasting. The C2C12 cell line is used as myoblasts or is differentiated in myotubes, sarcopenia is induced by dexamethasone (DEX). A Bonferroni significant difference procedure is used for a post hoc comparison. DEX toxicity (0.01–300 µM concentration range) is evaluated in myoblasts to measure cell viability and caspase 3 activation after 24 h and 48 h; cell incubation with 1 µM DEX for 48 h is chosen as optimal treatment for decreasing cell viability and increasing caspase 3 activity. R(+)LA or HMB significantly prevents DEX-induced cell mortality; the efficacy is improved when 100 µM R(+)LA is combined with 1 mM HMB. Regarding myoblasts, this combination significantly reduces DEX-evoked O_2_^−^ production and protein oxidative damage. During the early phase of myotube formation, the mixture preserves the number of myogenin-positive cells, whereas it completely prevents the DEX-dependent damage in a later phase of myotube differentiation (7 days), as evaluated by cell diameter and percentage of multinucleated cells. R(+)LA in association with HMB is suggested for sarcopenia therapy.

## 1. Introduction 

Sarcopenia is characterized by loss of skeletal muscle mass combined with decreased skeletal muscle quality (skeletal muscle performance per unit skeletal muscle mass) [1]. Age and several catabolic conditions such as starvation, diabetes, sepsis, inactivity and drugs (like glucocorticoids and chemotherapeutics) favor its development and strongly reduce quality of life, causing frailty [2]. Sarcopenia is considered a predictor of future mortality in middle-aged as well as older adults [3,4].

The mechanisms underlying the development of sarcopenia are multi-factorial and include the increase of protein breakdown [5,6], oxidative stress [7,8,9,10] and nervous alterations [11]. Protein degradation in skeletal muscle can be promoted by three different pathways: the ubiquitin–proteasomal pathway, the autophagic-lysosomal pathway and the calpain pathway. The ubiquitin–proteasomal pathway mediates the turnover of the majority of muscular protein; it is associated with the expression of the forkhead box O (FoxO) transcription factor that promotes the expression of muscle specific ubiquitin ligases such as atrophy gene-1/muscle atrophy F-box and muscle ring-finger protein 1 (MuRF1) [12,13], which mediate bond formation between protein and ubiquitin molecules [14]. The up-regulation of this pathway leads to muscle atrophy [15,16] and can be associated with the activation of apoptotic signals. The activation of caspase-3 in catabolic states yields a significant increase in protein degradation by the ubiquitin-proteasome system [17]. Indeed, caspase-3 and -8 inhibitors reduce the auto-phosphorylation of double-stranded RNA-dependent protein kinase whose activation induces an increase in protein breakdown in response to pro-inflammatory cytokines (such as TNF-α) and reactive oxygen species (ROS) production [18]. Circularly, sarcopenia favors oxidative stress. Regarding the case of muscle mass reduction, the function of the antioxidant enzymes catalase and glutathione peroxidase decline [9,10], and mitochondrial activity is altered resulting in an overproduction of free radicals [19,20]. ROS may trigger apoptosis [21].

Despite the knowledge of muscle mass regulation mechanisms gained lately, effective treatments for sarcopenia are lacking. Recently, nutrient supplementation with the branched-chain amino acid leucine and its metabolite β-hydroxy-β-methyl butyrate (HMB) emerged [22] as able to improve protein balance and reduce muscle wasting in cancer [23], acquired immunodeficiency syndrome (AIDS) [24], and aging [25]. This effect can be associated to HMB’s ability to attenuate the depression of protein synthesis induced by both a proteolysis-inducing factor and angiotensin II [26], and the induction of protein degradation in murine myotubes induced by a proteolysis-inducing factor [27]. Moreover, HMB has been shown to modulate the ubiquitin-proteasome system acting on the mammalian target of the rapamycin (mTOR) pathway [28,29] and to inhibit apoptosis induced by double-stranded RNA-dependent protein kinase [18]. Aiming to plan a potentially active synergistic association against sarcopenia, we conceive and test the association of HMB with the naturally occurring antioxidant compound lipoic acid (LA). LA is able to reduce mitochondrial dysfunction by acting as a cofactor in mitochondrial dehydrogenase complexes and, consequently, hinder ROS formation [30,31,32]. LA is widely employed as an adjuvant treatment of neuropathic pain and its ability to attenuate muscle damage/wasting has emerged as well [33,34]. LA has two optical isomers designated as R(+) and S(–). The natural enantiomer R(+)LA has been recently described as the eutomer, at least regarding antioxidant effects as well as neurovascular protection and pain relief [21,35]. The lysine salt of R(+)LA is quite stable and, therefore, useful in therapeutic applications.

The protective effects of HMB and R(+)LA, alone or in mixture, are evaluated in an in vitro model of DEX-induced sarcopenia, a model widely used by several research groups, both in vitro and in vivo, to study the mechanisms that cause sarcopenia [36,37,38,39]. Furthermore, weight loss, muscle wasting, and cachexia in patients are well-known side effects of glucocorticoid treatment, thus, the discovery of drugs to protect against protein degradation is an important research goal to investigate further [40,41].

## 2. Materials and Methods

### 2.1. Cell Cultures

C2C12 mouse skeletal myoblasts were obtained from American Type Culture Collection (Manassas, VA, USA), grown in Dulbecco’s Modified Eagle Medium (DMEM) (Euroclone, Milan, Italy) supplemented with 10% fetal bovine serum (FBS) (Euroclone, Milan, Italy), 100 U/mL penicillin and 100 µg/mL streptomycin, 200 mM L-glutamine (Life Technologies Italia, Milan, Italy) and maintained at 37 °C in a humidified atmosphere of 5% CO_2_. To induce myoblast differentiation into myotubes, C2C12 were cultured until reaching 80% of confluency, after which cells were shifted to a differentiation medium (DM: DMEM supplemented with 2% horse serum) for 72 h and for 7 days).

### 2.2. Pharmacological Treatments

Upon reaching confluence, myoblasts were plated in appropriate cell culture plates (Corning Costar, Milan, Italy) and treated with dexamethasone (DEX) (Merck, Milan, Italy) at different concentrations for 24 or 48 h. Lysine thioctate (R(+) lipoic acid-R(+)LA) and calcium β-hydroxy-β-methylbutyrate (HMB) (Barentz, Milan, Italy) were used either alone or in combination, in the presence or absence of DEX.

### 2.3. Cell Viability Assay

Myoblast cell viability was evaluated by the reduction of 3-(4,5-dimethylthiozol-2-yl)-2,5-diphenyltetrazolium bromide (MTT) (Merck, Milan, Italy) as an index of mitochondrial functionality. Cells were plated into 96-well cell culture plates (3 × 10^3^ cells/well). After 48 h, cells were treated with DEX (0.01, 0.03, 0.1, 1, 3, 10, 30, 100, 300 µM) for an additional 24 or 48 h. Myoblasts also were incubated with R(+)LA (1, 10, 30, 100, 300 µM) and HMB (1, 10, 30, 100, 300, 1000, 3000 µM) either alone or in combination, in the presence or absence of DEX (0.01, 0.03, 0.1, 1, 10 µM) for 48 h. All treatments were done in serum-free DMEM. After extensive washing, 1 mg/mL MTT was added into each well and incubated for 30 min at 37 °C. Then, the formazan crystals were dissolved in 150 mL dimethyl sulfoxide. The absorbance was measured at 550 nm. Experiments were performed in quadruplicate on at least three different cell batches.

### 2.4. Caspase-3 Activity

C2C12 myoblasts were plated into 6-well plates (5 × 10^4^ cells/well) and grown until confluent. Incubation, with increasing concentrations of DEX (0.01–100 μM) and four different mixtures of R(+)LA and HMB (30 μM and 100 μM, 30 μM and 300 μM, 100 μM and 300 μM, 100 μM and 1000 μM, respectively), was allowed for 48 h. Concurrently, 1 μM DEX was incubated with increasing concentrations of R(+)LA (1, 10, 30, 100, 300 µM) and HMB (30, 100, 300, 1000, 3000 µM), either alone or in combination, for 48 h. After treatment, cells were scraped into 100 μL of lysis buffer (200 mM tris-hydrogen chloride (Tris-HCl) buffer, pH 7.5, containing 2 M NaCl, 20 mM ethylenediaminetetraacetic acid (EDTA), and 0.2% Triton X–100) (Merck, Milan, Italy). Fifty microlitres of the supernatant was incubated with 25 μM of the fluorogenic peptide caspase substrate, rhodamine 110 bis (N-CBZ-L-aspartyl-L-glutamyl-L-valyl-L-aspartic acid amide) (Molecular Probes, Milan, Italy), at 25 °C for 30 min. The amount of cleaved substrate was measured in a 96-well plate fluorescence spectrometer (FlexStation 3, Molecular Devices; excitation at 496 nm and emission at 520 nm).

### 2.5. Superoxide Dismutase-Inhibitable Superoxide Anion (O_2_^−^) Production Evaluated by Cytochrome C Assay

C2C12 myoblasts were plated in six-well plates (5 × 10^4^ cells/well) and grown until confluent. Cells were then incubated with increasing concentrations of DEX (0.01, 0.03, 0.1, 0.3, 1, 10, 100 µM) and with 1 µM DEX in the presence or absence of 100 µM R(+)LA, 1 mM HMB and the mixture 100 µM R(+)LA–1 mM HMB in serum-free DMEM containing cytochrome C (1 mg/mL; Sigma-Aldrich, Milan, Italy) for 4 h at 37 °C. Nonspecific cytochrome C reduction was evaluated by carrying out tests in the presence of bovine superoxide dismutase (SOD; Sigma-Aldrich, Milan, Italy) (300 mU/m). The supernatants were collected and the optical density was measured at 550 nm. After the nonspecific absorbance was subtracted, the SOD-inhibitable O_2_^−^ amount was calculated using an extinction coefficient of 2.1 × 10^4^ M^−1^ × cm^−1^ and expressed as µM/mg protein/4 h. The 4 h incubation interval was chosen on the basis of preliminary experiments which showed poor reliability for longer cytochrome C exposure to the cellular environment.

### 2.6. Protein Extraction and Quantification

C2C12 myoblasts were plated in a 25 cm^2^ cell culture flask (10^5^ cells/flask) and grown until confluent. To induce the differentiation, C2C12 myotubes were plated in a 25 cm^2^ cell culture flask (7 × 10^4^ cells/flask) and grown until 80% confluent. After that, cells were shifted to DM for 7 days. The DM was replaced every 2 days. Carbonylated proteins were evaluated after a 48 h incubation with 1 µM DEX with or without 100 µM R(+)LA, 1 mM HMB and the mixture 100 µM R(+)LA-1 mM HMB. After incubation, myotube cell cultures were washed once with phosphate-buffered saline (PBS) (Euroclone, Milan, Italy) and scraped onto ice with a lysis buffer containing 50 mM Tris-HCl pH 8.0, 150 mM NaCl, 1 mM EDTA, 0.5% Triton X-100, and Complete Protease Inhibitor (Roche, Milan, Italy). Suspensions were then collected, subjected to a freeze-thaw cycle, and centrifuged at 13,000× *g* for 10 min at 4 °C. Protein concentrations were quantified by a bicinchoninic acid assay.

### 2.7. Carbonylated Protein Evaluation

Protein carbonylation was evaluated both in C2C12 myoblasts and C2C12 differentiated myotubes. After extraction, 20 μg samples of protein were denatured in 6% sodium dodecyl sulfate (Merck, Milan, Italy) and derivatized with 10 mM 2,4-dinitrophenyl hydrazine (DNPH) (Merck, Milan, Italy) for 15 min at room temperature. Samples were separated on a 12% sodium dodecyl sulfate-polyacrylamide gel by electrophoresis (SDS-PAGE) and transferred onto nitrocellulose membranes (Bio-Rad, Milan, Italy). Membranes were blocked with 1% bovine serum albumin (BSA) (Merck, Milan, Italy) in PBS containing 0.1% Tween 20 (PBST) and then probed overnight with specific primary antibody versus DNPH (1:5000 in PBST/1% BSA). After being washed with PBST, the membranes were incubated for 1 h in PBST containing the appropriate horseradish peroxidase-conjugated secondary anti-rabbit (1:5000; Cell Signaling, Danvers, MA, USA) and again washed. Enhanced chemiluminescence (ECL) (Pierce, Rockford, IL, USA) was used to visualize the peroxidase-coated bands. Densitometric analysis was performed using the Scion Image analysis software. Regarding each experiment, the density of all bands shown in a lane was reported as the mean. Glyceraldehyde-3-phosphate dehydrogenase (GAPDH) normalization was performed [42].

### 2.8. Western Immunoblot Analysis

50 μg of each sample were resolved with 10% SDS-PAGE and transferred onto nitrocellulose membranes (Bio-Rad, Milan, Italy). Membranes were blocked with 5% nonfat dry milk in PBST and then probed overnight at 4 °C with primary antibody-specific versus muscle ring-finger protein-1 (MuRF1) (1:1000; 40 kDa; Santa Cruz, CA, USA), forkhead in rhabdomyosarcoma (FKHR) (1:1000; 80 kDa; Santa Cruz, CA, USA) or GAPDH (1:5000; 37 kDa; Santa Cruz, CA, USA). Membranes were then incubated for 1 h in PBST containing the appropriate horseradish peroxidase-conjugated secondary anti-rabbit (1:5000; Cell Signaling), anti-goat (1:5000; Merck, Milan, Italy) or anti-mouse antibody (1:2000; Santa Cruz, CA, USA). ECL was used to visualize the peroxidase-coated bands. Densitometric analysis was performed using the “ImageJ” analysis software (ImageJ, NIH, Bethesda, MD, USA) and results were normalized to α-Tubulin immunoreactivity as an internal control. Values were reported as percentages in comparison to the control, which was arbitrarily fixed at 100%.

### 2.9. Immunofluorescence

C2C12 myoblasts were plated into coverslips (5 × 10^3^ cells/slice) and grown until confluent. After that, cells were exposed to DM for 24 h and, subsequently, subjected to pharmacological treatment for 48 h in DM. Cells were treated with 1 µM DEX in the presence or absence of 100 µM R(+)LA, 1 mM HMB or the mixture of both 100 µM R(+)LA–1 mM HMB and then fixed in 4% buffered paraformaldehyde for 10 min at room temperature. Fixed cells were permeabilized with PBS containing 0.1% Triton X-100 for 10 min and then incubated with a blocking solution containing 0.5% BSA and 3% glycerol in PBS for 30 min. After blocking, cells were incubated at 4 °C overnight with mouse monoclonal anti-myogenin (1:50; Santa Cruz, CA, USA). To reveal the immunostaining, the cells were incubated with goat anti-mouse Alexa Fluor 568-conjugated IgG (1:200; Life Technologies, Italy) for 1 h at room temperature. Negative controls were carried out by replacing the primary antibody with non-immune mouse serum; cross-reactivity of the secondary antibody was tested in control experiments in which the primary antibody was omitted. During some experiments, counterstaining was performed with either TRITC-labeled phalloidin (1:40; Life Technologies, Italy) to reveal filamentous actin and 4′,6-diamidine-2′-phenylindole dihydrochloride (DAPI) (1:2000; Merck, Milan, Italy) to reveal nuclei. After washing, the coverslips containing the immunolabeled cells were mounted with the mounting medium ProLong (Life Technologies, Milan, Italy) and observed under a motorized Leica DM6000B microscope equipped with a DFC350FX camera (Leica, Mannheim, Germany). Quantitative analysis of myogenin-positive cells was performed by collecting at least three independent fields through a 20X 0.5NA objective. Myogenin-positive cells were counted in 72 h differentiated myotubes using the ‘‘cell counter’’ plugin of ImageJ. The myogenin signal in immunostained sections was quantified using FIJI software (distributed by ImageJ, NIH, Bethesda, MD, USA) by automatic thresholding images with the aid of the ‘‘Moments’’ algorithm, which we found to provide the most consistent pattern recognition across all acquired images. Results were expressed as a percentage calculated by the ratio between the number of myogenin-positive cells and the total cells identified by DAPI (100%).

### 2.10. Morphologic Evaluations

C2C12 myotubes were plated on coverslips (5 × 10^3^ cells/slice) and grown until confluent. After that, cells were shifted to DM for 7 days. The DM was replaced every 2 days. Pharmacological treatments were performed by incubating seven day-differentiated myotubes with 1 µM DEX in the presence or absence of 100 µM R(+)LA, 1 mM HMB, or the mixture of 100 µM R(+)LA-1 mM HMB for 48 h. After that, fixed cells were immunolabeled with either TRITC-labeled phalloidin (1:40; Life Technologies, Italy) to reveal filamentous actin or DAPI (1:2000) to reveal nuclei. The evaluation of myotube diameters, number of nuclei per myotube and multinucleated cells were done using image analyzing software (ImageJ 1.48). Morphometric analysis was performed by collecting at least three independent fields through a 20X 0.5NA objective, and micrographs to be analized were taken using a motorized Leica DM6000B microscope equipped with a DFC350FX camera (Leica, Mannheim, Germany).

### 2.11. Statistical Analysis

Results were expressed as mean ± S.E.M. and analysis of variance (ANOVA) was performed. A Bonferroni significant difference procedure was used as a post hoc comparison. All assessments were made by researchers blinded to cell treatments. Data were analyzed using the “Origin 8.1” software (OriginLab, Northampton, MA, USA).

### 3. Results

#### 3.1. C2C12 Myoblasts

Dexamethasone (DEX) decreased the viability of C2C12 myoblasts after 24 h incubation, taking effect starting from 0.1 μM (Table 1). Reduction by about 30–40% was reached using 1 μM DEX. The same concentration lowered cell viability by 50% at 48 h. Higher concentrations did not increase the cell mortality rate (after both 24 and 48 h: Table 1). Choosing 1 μM DEX for 48 h as standard damage, C2C12 myoblasts were treated with R(+) stereoisomer of lipoic acid (R(+)LA) and β-hydroxy-β-methyl butyrate (HMB) for testing protective properties. Shown in Figure 1, R(+)LA (100 and 300 μM) prevented DEX-induced cell damage, increasing viability from 50% (DEX) to about 75% (300 μM R(+)LA). HMB showed a significant effect starting from 1 mM, and a complete prevention of mortality was observed at 3 mM (Figure 1). Considering concentrations of 100 μM LA and 1 mM HMB, full restoration of cellular viability was observed (Figure 1), suggesting synergism.

Effects of R(+)LA and HMB on the viability of C2C12 treated with DEX in the dose range 0.01–10 μM for 48 h is shown in Appendix A.

During the absence of DEX, R(+)LA (1–300 μM) and HMB (1 μM–1 mM) alone did not alter myoblast viability, whereas 3 mM HMB and the mixture R(+)LA + HMB were able to increase C2C12 viability. Particularly, 100 μM R(+)LA/1 mM HMB enhanced viability by about 70% (Appendix A). Shown in Table 2, DEX induced the apoptotic processes promoting caspase-3 activity. Characteristically, the enzymatic activity was increased by about 70% in the concentration range of corticosteroid 0.01–1 μM, while 10 and 100 μM were no more able to induce caspase-3 activation (Table 2), suggesting that higher concentrations may overcome the apoptotic phenomena evoking toxicity signals that lead to cell mortality (shown also at these concentrations, Table 1) probably by necrotic mechanisms.

The pro-apoptotic activity induced by 1 μM DEX (48 h) was prevented by 1 and 3 mM HMB (Figure 2). A similar protective effect was induced by the mixture 100 μM R(+)LA/300 μM HMB (Figure 2).

Regarding the absence of DEX, the basal activity of caspase-3 was not modified neither by R(+)LA nor by HMB (Appendix A).

The treatment of C2C12 myoblasts with DEX evoked an oxidative derangement. O_2_^−^ levels increased concentration-dependence after 48 h of treatment (Appendix A), inducing alterations of macromolecules like protein carbonylation (Figure 3B). The combination of R(+)LA (100 μM) and HMB (1 mM) was necessary to significantly prevent both O_2_^−^ increase (Figure 3A) and protein carbonylation (Figure 3B) promoted by DEX (1 μM, 48 h).

#### 3.2. C2C12 Myotubes

DEX (1 μM, 48 h, added during the differentiation period between 24 and 72 h) reduced the number of myogenin-positive cells by about 30%. The co-treatment with 100 μM R(+)LA or 1 mM HMB, alone or in mixture, restored myogenin expression (Figure 4).

The complete maturation of C2C12 myotubes was observed after 7 days of differentiation (F-actin staining, Figure 5) (cells showing two or more nuclei, as according to Lee [43]).

The subsequent treatment with DEX (1 μM, 48 h) altered the myotube morphology, reducing the myotube diameter (Figure 6A) and the number of multinucleated myotubes (Figure 6B) by about 36% and 27%, respectively. Concerning the presence of 100 μM R(+)LA or 1 mM HMB, alone or in mixture, morphological alterations were prevented (Figure 5 and Figure 6). The number of nuclei per myotube was not modified by DEX or R(+)LA/HMB. Occurring on day 7 of differentiation, myogenin was no longer detectable (data not shown).

Shown in Figure 7, DEX also evoked oxidative damage in myotubes. Carbonylated protein expression was increased up to two fold (Figure 7A). Furthermore, muscle ring-finger protein 1 (MuRF1) and forkhead box O (FoxO) expression levels were increased up to three fold (Figure 7B,C). The mixture of 100 μM R(+)LA/1 mM HMB fully prevented protein oxidation, on the contrary, it did not modify the alterations induced by the corticosteroid on MuRF1 and FoxO (Figure 7).

### 4. Discussion

The described results show the synergistic effect of R(+) stereoisomer of lipoic acid (R(+)LA) and β-hydroxy-β-methyl butyrate (HMB) against the dexamethasone (DEX)-dependent damage of myoblast- and myotube-cell cultures. The optimal ratio between compounds has been defined.

Catabolic effects of glucocorticoids, including DEX, on muscle protein metabolism are well known. It is generally accepted that glucocorticoids are able to induce muscular atrophy, inhibiting muscle protein synthesis and stimulating muscle protein breakdown in both in vivo [44,45,46,47] and in vitro models [48,49,50]. The myotoxicity of DEX depends on cell type and is concentration-dependent with values between 10 nM and 100 μM [15,48,49,51,52]. Our result of a threshold value of 1 μM agrees with other reports regarding viability, caspase-3 activity and muscle ring-finger protein 1 (MuRF1) and forkhead box O (FoxO) expression [50,53], both of which play a pivotal role in the ubiquitin–proteasome-dependent muscle atrophy induced by glucocorticoids [54].

Found in this condition, both compounds R(+)LA and HMB, were able, in a concentration-dependent manner, to protect C2C12 muscle cells from DEX-induced damage; moreover, the combination promoted a synergistic effect regarding several parameters. Regarding myoblasts, the mixture R(+)LA/HMB (100 µM and 1 mM, respectively) enhanced cell viability in the control condition as well, suggesting a proliferative effect without decreasing the basal caspase-3 activity. Conversely, this effect on the basal condition agrees with the antioxidant potential of R(+)LA of reducing the oxidative stress produced by cell growth favoring proliferation, as well as by the nutrient supplementation due to HMB as a metabolite of the branched-chain amino acid leucine [22].

The mixture was more effective and powerful in comparison to single compounds in protecting from the toxicity induced by DEX, both as a mortality and apoptosis trigger. It is interesting to note that the combination R(+)LA/HMB was the only treatment able to significantly reduce the DEX-dependent redox imbalance by reducing the concentration of O_2_^−^ and the level of carbonylated proteins, an oxidative modification due to the introduction of carbonyl groups into protein side chains by a site-specific mechanism [42] leading to decreased functionality. Furthermore, the protective effects of the combination R(+)LA/HMB also were highlighted in myotubes. During myotube differentiation from myoblast cultures, transcription factors of the myogenic differentiation gene family, including myogenin, play a pivotal role in the regulation of the fusion of myoblasts to form myotubes [55,56,57]. Agreeing with results from others [13,58] we found that DEX decreased the expression of myogenin in C2C12 myotubes. Furthermore, the diameter and the formation of mature myotubes were reduced. R(+)LA and HMB, alone and in combination, were able to counteract the decrease in myogenin-positive cell numbers induced by DEX, as well as to restore alterations in myotube diameters. Particularly, the co-treatment R(+)LA/HMB increased the number of multinucleated cells over the value of control cells. Again, the concomitant presence of both compounds was the only treatment able to significantly prevent the oxidative damage of myotube proteins.

Glucocorticoids lead to mitochondrial dysfunction and this has been linked to excessive reactive oxygen species (ROS) production [59,60]. Among other effects, both compounds R(+)LA and HMB possess antioxidant activities which involve the reduction of ROS by scavenging properties or by restoring the glutathione redox state due to the elevation of intracellular cysteine levels [61]. These mechanisms are related to the attenuation of muscle wasting [62,63]. R(+)LA is the centerpiece of the pyruvate dehydrogenase complex, which functionally links glycolysis in the cytoplasm to oxidative phosphorylation in the mitochondria; its antioxidant properties are largely reported [61,64]. Lipoic acid (LA) can actively counter various forms of oxidative stress; it prevents the damage caused by oxygen free radicals thanks to its ability to cross cell membranes very quickly, and to act as an antioxidant in both lipid and aqueous phases [65,66,67]. LA biological activity is particularly referable to the natural R dextrorotatory enantiomer [21,35]. Only R(+)LA is synthesized by cells and works as a cofactor for some critical mitochondrial enzymes: pyruvate dehydrogenase, branched-chain α-keto-acid dehydrogenase, and α-ketoglutarate dehydrogenase [68]. Recently, other peculiar effects of LA against muscle wasting have emerged. LA was effective against diabetic myopathy by reducing morphological alterations in slow and fast rat skeletal muscles [69] and preserved muscle mass in diabetic rats by upregulating the AMPK/SIRT1/PGC-1α (5’AMP-activated protein kinase/ Sirtuin 1/ peroxisome proliferator-activated receptor γ coactivator 1 α) and AKT/mTOR/p70S6K (protein kinase B/ mammalian target of rapamycin/ p70S6 kinase) signaling pathways [34]. Particularly, the role of PGC-1α in LA-mediated muscle protection was recently confirmed in a rat model of DEX-dependent sarcopenia [70].

Regarding HMB, it is metabolized in the cytosol joining a coenzyme A molecule, the resulting compound is converted in β-hydroxy-β-methylglutaryl coenzyme A (HMG-CoA), the primary metabolite of HMB in the body. HMG-CoA is then reducted into mevalonic acid, one of the precursors for the synthesis of cholesterol, which is involved in the integrity of the myocyte cell membrane [71].

Moreover, HMB showed direct antioxidant effects. Regarding murine myotubes, HMB reduced ROS formation induced by lipopolysaccharides [72]; concerning humans, it improves the dynamics of mitochondria, an intracellular organelle strongly affected by sarcopenia-like conditions which induce energy stress and elevated ROS production [73]. Conversely, HMB presents a multifactorial pharmacodynamic mechanism of action justifying its efficacy in muscle cell protection and muscular performance enhancement. As a metabolite of the branched-chain amino acid leucine, several studies recently reported that HMB supplementation represents a strategy for increasing power and muscle hypertrophy in healthy, trained subjects [74]. HMB reduces protein breakdown acting on the mTOR pathway and by inhibiting the ubiquitin-proteasome proteolytic pathway [28,29,75], further, it stimulates protein synthesis as well as myogenic differentiation and survival via the PI3K (phosphoinositide 3-kinase)/AKT pathway [76]. To note, these positive profiles appear to be strongly related to the use of a correct dosage [75], since low dosage resulted in no effective treatments [77]. To summarise, the beneficial effects of LA and HMB in counteracting DEX-induced myotoxicity are due to their antioxidant property, their activation of the mTOR cascade, and their inhibition of the proteasome pathway.

### 5. Conclusions

These results highlight the protective effects of R(+)LA combined with HMB in myoblast- and myotube-cultures damaged by DEX. In vivo preclinical studies are necessary for the clinical employment of the combination, based on the extensive use of both individual compounds in humans and their good safety profile. These data offer a rationale to candidate the mixture as a therapeutic option for sarcopenia treatment.

## Figures and Tables

**Figure 1 molecules-25-02117-f001:**
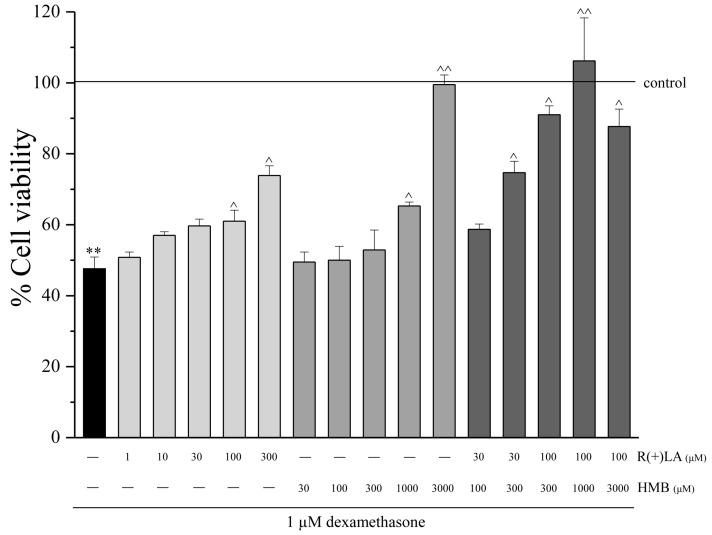
Viability of C2C12 myoblasts. C2C12 myoblasts were treated with 1 μM DEX in the absence or presence of increasing concentrations of R(+)LA (1–300 μM) or HMB (30 μM–3 mM), or the combination of both for 48 h. Cell viability was measured by 3-(4,5-dimethylthiozol-2-yl)-2,5-diphenyltetrazolium bromide (MTT) assay. The control condition was arbitrarily set as 100% (black line) and values are expressed as the mean ± S.E.M. of three experiments. ** *p* < 0.01 versus control; ^ *p* < 0.05 and ^^ *p* < 0.01 versus DEX treatment in the absence of R(+)LA and HMB.

**Figure 2 molecules-25-02117-f002:**
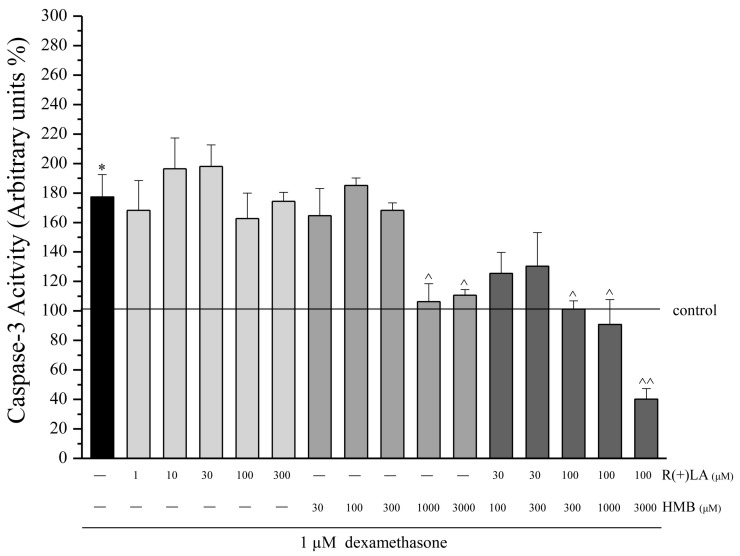
Caspase-3 activity percentage in C2C12 myoblasts. C2C12 myoblasts were treated with 1 μM DEX in the absence or presence of increasing concentrations of R(+)LA (1–300 μM) or HMB (30 μM–3 mM), or the combination of both, for 48 h. Caspase-3 activity was measured by a fluorescence assay. The control condition was arbitrarily set as 100% (black line) and values are expressed as the mean ± S.E.M. of 3 experiments. * *p* < 0.05 versus control; ^ *p* < 0.05, ^^ *p* < 0.01 versus DEX treatment in the absence of R(+)LA and HMB.

**Figure 3 molecules-25-02117-f003:**
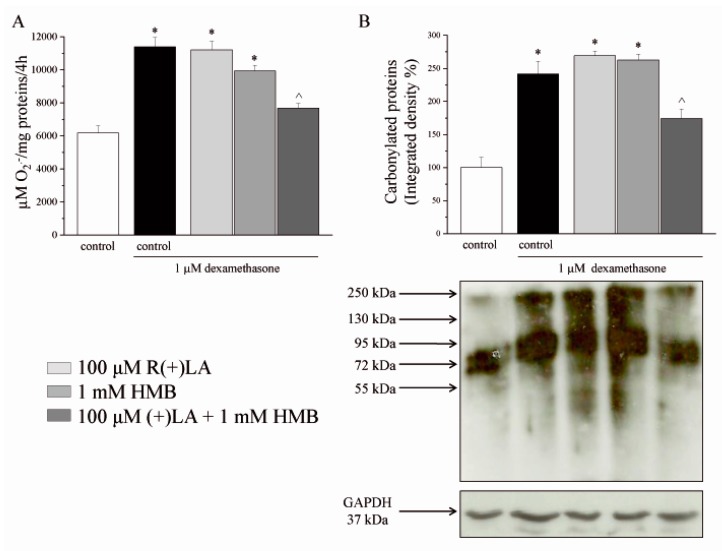
Oxidative stress in C2C12 myoblasts. (**A**) O_2_^−^ levels. C2C12 myoblasts were exposed to 1 μM DEX for 4 h in the absence or in the presence of 100 μM R(+)LA or 1 mM HMB or the combination of both. O_2_^−^ concentration was evaluated by cytochrome C assay. The non-specific absorbance was measured in the presence of superoxide dismutase (SOD) (300 mU/mL) and subtracted from the total value. Values are expressed as µM/mg protein/4 h. Bars represent the mean ± S.E.M. of 3 experiments. (**B**) Protein carbonylation. C2C12 myoblasts were treated for 48 h with 1 μM DEX in the absence or presence of 100 μM R(+)LA or 1 mM HMB or the combination of both. Western blot analysis was performed on cell homogenates using a specific antibody against DNPH. Densitometric analysis (top) and representative immunoblot (bottom) are shown. GAPDH normalization was performed for each sample. Values are expressed as the mean ± S.E.M. percent of control performed in 3 different experiments. * *p* < 0.05 versus control (in the absence of DEX) and ^ *p* < 0.05 versus DEX treatment in the absence of R(+)LA and HMB.

**Figure 4 molecules-25-02117-f004:**
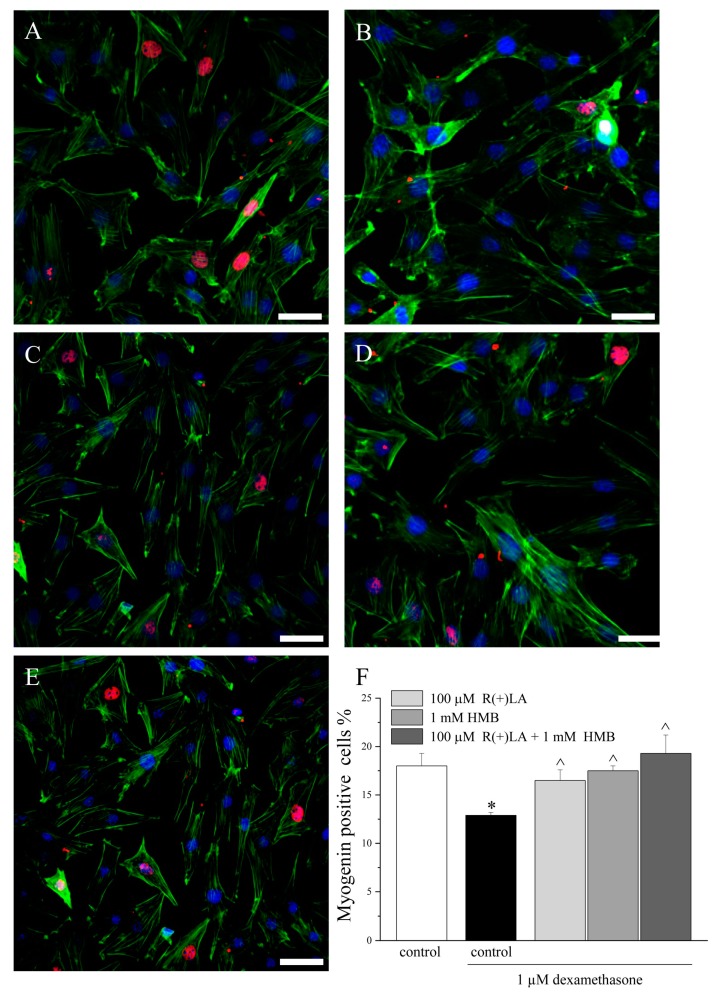
Myogenin staining of C2C12 cells after 72 h differentiation. C2C12 myoblasts were cultured for 24 h in differentiation medium (DM) and then incubated for 48 h as follows: with DM (**A**); with 1 μM DEX (**B**); with 1 μM DEX and 100 μM R(+)LA (**C**); with 1 μM DEX and 1 mM HMB (**D**); with 1 μM DEX, 100 μM R(+)LA and 1 mM HMB (**E**). The cells were then fixed and stained with Alexa488-conjugated phalloidin to evidence F-actin organization and a specific antibody that recognized myogenin (red). Nuclei were stained with 4′,6-Diamidine-2′-phenylindole dihydrochloride (DAPI) (blue). (**F**) Quantitative analysis of the percentage of myogenin-positive cells. The percentage was calculated by the ratio between the number of myogenin-positive cells (red) and the total cells identified by DAPI (100%). Scale bar (white line): 50 μm. Values are expressed as the mean ± S.E.M. of 3 different experiments. * *p* < 0.05 versus control (in the absence of DEX) and ^ *p* < 0.05 versus DEX treatment in the absence of R(+)LA and HMB.

**Figure 5 molecules-25-02117-f005:**
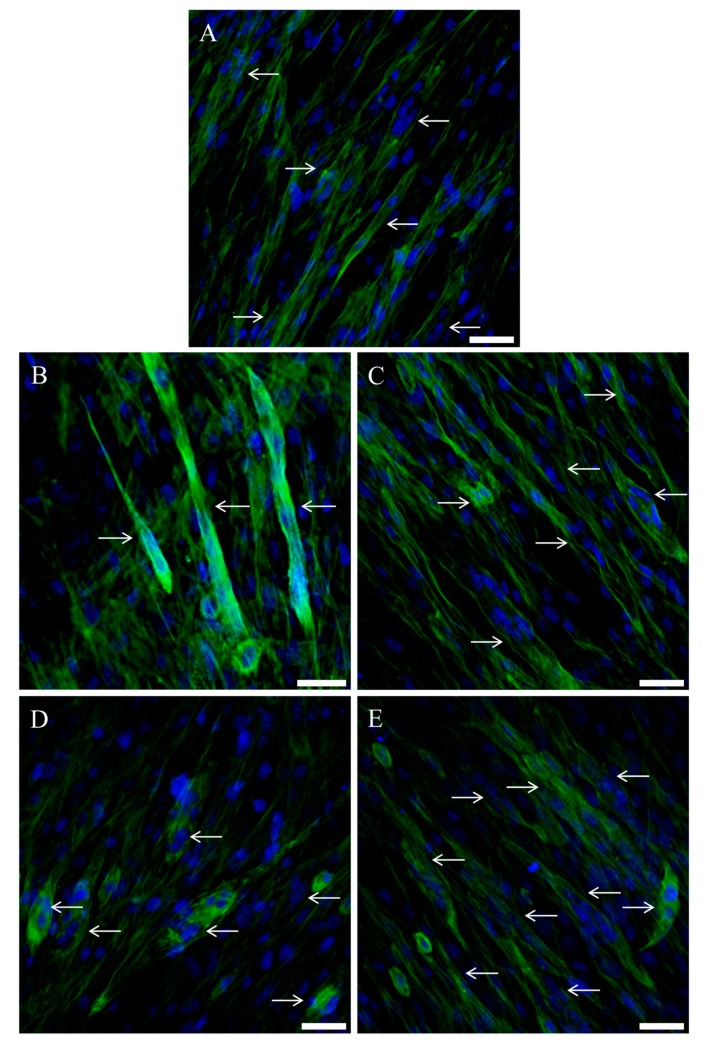
Morphological analysis of C2C12 myotubes. C2C12 myoblasts were cultured in DM for 7 days. The cultures were then incubated for 48 h with DM (**A**); with 1 μM DEX (**B**); with 1 μM DEX and 100 μM R(+)LA (**C**); with 1 μM DEX and 1 mM HMB (**D**); with 1 μM DEX, 100 μM R(+)LA and 1 mM HMB (E). Cells were then fixed and stained with Alexa488-conjugated phalloidin to evidence F-actin organization. Nuclei were stained with DAPI (blue). Scale bar (white line): 50 μm.

**Figure 6 molecules-25-02117-f006:**
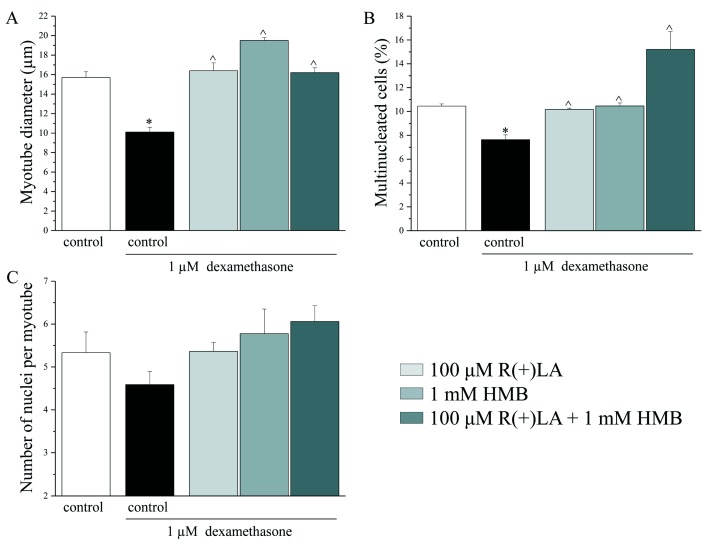
Morphometric analysis of C2C12 myotubes. C2C12 myoblasts were cultured in DM for 7 days. The cultures were then incubated for 48 h with 1 μM DEX in the absence or presence of 100 μM R(+)LA or 1 mM HMB, or the combination of both. After phalloidin staining, morphometric analyses were performed to provide: the measurement of the myotube diameter (**A**), the multinucleated cell percentage (**B**) (determined as the ratio percent between the number of multinucleated cells and the number of total cells considered as 100%), and the number of nuclei per myotube (**C**). The analyses were conducted in at least 3 random fields in each experimental group. Values are expressed as the mean ± S.E.M. of three different experiments. * *p* < 0.05 versus control (in the absence of DEX) and ^ *p* < 0.05 versus DEX treatment in the absence of R(+)LA and HMB.

**Figure 7 molecules-25-02117-f007:**
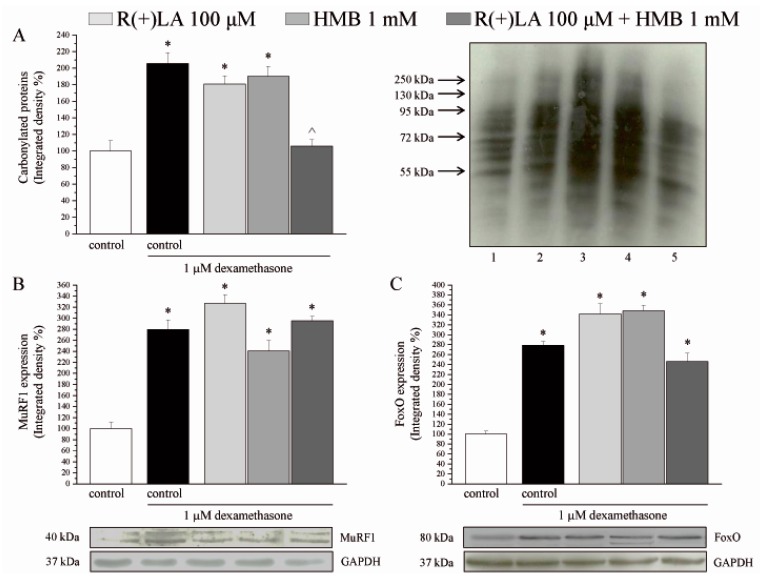
Protein expression levels of carbonylated proteins, MuRF1 and FoxO in C2C12 myotubes. Myotubes (7 day differentiation) were incubated for 48 h with 1 μM DEX in the absence or presence of 100 μM R(+)LA or 1 mM HMB, or the mixture of both. (**A**) Protein carbonylation. Densitometric analysis (left) and representative immunoblot (right) are shown. (**B**) MuRF1 protein expression levels. Densitometric analysis (top) and representative immunoblot (bottom) are shown. (**C**) FoxO expression levels in C2C12 myotubes. Densitometric analysis (top) and representative immunoblot (bottom) are shown. Alpha-Tubulin normalization was performed for each sample. Values are expressed as the mean ± S.E.M. percent of control performed in 3 different experiments. * *p* < 0.05 versus control (in the absence of DEX) and ^ *p* < 0.05 versus DEX treatment in the absence of R(+)LA and HMB.

**Table 1 molecules-25-02117-t001:** Viability of C2C12 myoblasts.

Cell Viability %			
Dexamethasone Concentration (μM)	Incubation	
	24 h	48 h
0	100.0 ± 0.9	100 ± 2.4
0.01	87.32 ± 4.5	69.1 ± 1.8**
0.03	85.34 ± 6.7	62.4 ± 4.6**
0.1	68.9 ± 1.8**	58.9 ± 4.3**
1	65.4 ± 1.7**	47.6 ± 3.3**
3	62.2 ± 1.9**	56.1 ± 5.0**
10	63.3 ± 1.8**	52.3 ± 2.3**
30	62.5 ± 1.8**	53.5 ± 6.7**
100	61.3 ± 0.4**	52.4 ± 3.8**
300	59.5 ± 0.6**	54.2 ± 8.1**

Cell viability of C2C12 myoblasts treated with increasing concentrations of DEX (0.01–300 μM) for 24 h or 48 h. The control condition was arbitrarily set as 100% and values are expressed as the mean ± S.E.M. of 3 experiments. ** *p* < 0.01 versus control (0 µM).

**Table 2 molecules-25-02117-t002:** Caspase-3 activity in C2C12 myoblast.

Caspase-3 Activity %	
Dexamethasone Concentration (μM)	48 h Incubation
0	100 ± 3.4
0.01	160.6 ± 20.1*
0.1	171.7 ± 14.9*
0.3	170.3 ± 16.8*
1	177.3 ± 15.2*
10	106.2 ± 4.4
100	94.5 ± 5.9

Caspase-3 activity of C2C12 myoblasts treated with increasing concentrations of DEX (0.01–100 μM) for 48 h. The control condition was arbitrarily set as 100% and values are expressed as the mean ± S.E.M. of three experiments. * *p* < 0.05 versus control (0 µM).

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
