# Peer review of "Effects of the Combination of β-Hydroxy-β-Methyl Butyrate and R(+) Lipoic Acid in a Cellular Model of Sarcopenia"

_molecules, 2020, doi:10.3390/molecules25092117_

Round 1

Reviewer 1 Report

Understanding the molecular mechanisms regulating sarcopenia is of most importance for successful pharmaceutical intervention to protect muscle cells. The presented study investigates pharmacological profile of the leucine metabolite β-Hydroxy-β-methyl butyrate (HMB) in combination with the natural R(+) stereoisomer of lipoic acid (R(+)LA) in a cellular model of muscle wasting, mouse myoblast cell line C2C12 treated with dexamethasone. Although initial data on protective effect of the tested compounds alone and in combination on viability of damaged cells are very interesting and promising, the manuscript is largely descriptive and does not include mechanistical study. In addition, this study lacks novelty and originality; conclusions are overstated.

Major concerns:

1.      As mentioned above, presented study lacks comprehensive mechanistical study. Authors propose that HMB and R(+)LA increase viability of dexamethasone-treated C2C12 cells by protecting from dexamethasone-induced apoptosis due to a decrease in Caspase-3 activity as shown in Fig. 2. However, the mechanism of this decrease is not investigated.

2.      The other proposed direction of the protective effects of the tested compounds involves oxidative stress as shown (indirectly and limited) in Fig.3. However, the mechanism of this observed effect is not investigated. Oxidative stress does not include only superoxide production and increase in carbonylated proteins.

3.      Presented study is incoherent. There are several observations pointing to a possible protective effect of the tested compounds alone and in combination; however, the connection between these pieces of data is not investigated.    

Other concerns:

1.      Fig. 1 shows decrease in viability of dexamethasone-damaged cells treated with the combination of 0.1 mM R(+)LA and 3 mM HMB compared to protective effects of the lower concentrations of these compounds. Table 2 shows significantly increased viability of undamaged cells treated the same combination of the tested compounds. These two facts need explanation, since they point to a possibility of 2 very different mechanisms involved.

2.      In Tables 3, 4 and Fig. 2, Fig. 3B - data should be presented as fold-change of the control. Presenting data as % of control and setting control as 100% is misleading, since, for ex., in Fig. 2, control cells have 100% caspase-3 activity.

3.      In order to claim that protective effect of the tested compounds involves decreased apoptosis, at least 2 methods for apoptosis assessment should be presented. In addition to caspase-3 activity, either morphological features of the cells or Annexin 5 staining (or some other proof) should be shown.   

4.      In Fig. 3A and B; Fig. 7 - Y axis should start at 0.

5.      Figs. 4, 5, 6 - For myogenin staining, morphological and morphometric analyses of the cells, confocal microscopy should be used; especially for the quantitative aspects. It is also not clear whether the cells in Fig. 4F and Fig. 6B were counted per HPF. In addition, Fig 4 and Fig. 6 legends state that values are expressed as percent of control, while control is obviously not 100% (Fig. 4F, Fig. 6B).

6.      There is no justification or explanation for the assessment of MuRF1 and FoxO protein levels (Fig. 7).

7.      Discussion is very limited and not mostly connected with the results presented in the manuscript.

Reviewer 2 Report

Reviewer Recommendation and Comments for Manuscript # Molecules-751449

Effects of the combination of β-Hydroxy-β-Methyl Butyrate and R(+) Lipoic Acid in a cellular model of sarcopenia.

Comments

These are my general/specific comments:

GENERAL:  This original research manuscript concerns the effectiveness of a pharmacological/nutritional supplementation alone or in combination for the treatment of muscle wasting in an in vitro model of dexamethasone-induced sarcopenia; and, indeed, offers to bridge a gap in the literature that is a critical topic area of research and warrants examination such as the investigators have provided. Overall, this manuscript has only minor editorial issues and was a delight to critically review.  Indeed, the authors’ interpretation of the existing literature is acceptable as the manuscript logically interprets and supports the available applicable experimental data with relevant findings of their own. Further, the authors do address issues related to various efficacy issues as well as the paucity of available interventions for the focused application of such interventions on the sarcopenic state; however, an expanded explanation of the practicality of their intervention(s) in terms of the relevance of an optimal (or at minimum, a validated) translational and/or applicable in vivo approach would have immediate practical implications and should be included. This would be best served and included in a “practical application” paragraph in the Discussion section. Importantly, the authors must justify (or provide feasible evidence) or clarify the use of proper control group(s) utilized in their current investigation.  Even though there are multiple points that should be addressed concerning the existing manuscript, the authors should be able to address these points for full consideration.

SPECIFIC:

TITLE

None

ABSTRACT

None

INTRODUCTION

  1. Page 1; Paragraphs 1-2; Lines 1-7: The authors should be more specific in detailing/defining decrements in age-associated “strength and physical performance” decrements as it has been reported/established as “dynapenia” - see references: 
  • Clark, B.C.; Manini, T.M. Sarcopenia6= Dynapenia. J. Gerontol. Ser. A Biol. Sci. Med. Sci. 2008, 63, 829–834.
  • Clark, B.C.; Manini, T.M. What is dynapenia? Nutrition 2012, 28, 495–503.
  • Baker, BA. Funct. Morphol. Kinesiol. 2018, 3(2), 36; https://doi.org/10.3390/jfmk3020036

Further, the authors should consider the suggestion and references cited above in  terms of discussing the age-specific reduction in physical performance in general and  strength/power specifically with regards to dynapenia, as well as it’s impact on  sarcopenia (and vice- versa).

If the authors believe that this concept of distinguishing the accepted (and published)  evidence-based findings are not relevant and wish to keep with a more historical  definition of sarcopenia, then the authors must include an expanded  discussion/premise for functional (i.e. Skeletal muscle performance metrics - strength,  power, work capacity) impact owing to or impacting the sarcopenic state.

  1. Page 1; Paragraph 1; Line3: Add “and” following “development” and change  “reducing” to “reduces”.
  2. Page 2; Paragraph 1; Line 6: Delete “to” following “yields”.
  3. Page 2; Paragraph 2; Line 3: Write-out the word HMB, prior to using its  abreviation/acronym.
  4. Page 2; Paragraph 2; Line 5: Change “HMB” to “HMB’s”.
  5. Page 2; Paragraph 2; Line 12: Add “by” following “dysfunction”.
  6. Page 2; Paragraph 2: Line 17: Change “...allows a stability enough for therapeutic  purposes” to “...exhibits a stability for use as a therapeutic intervention”.

MATERIALS and METHODS

  1. Page 2; Paragraph 2; Line 1: Change “reached” to “reaching”
  2. Page 2; Paragraph 2: The authors should justify (or consider including, if a justification is not feasible) why they did not include a zero-dose control group for dexamethasone. It is not justifiable to accept the 0.01 uM dose as a 0 uM dose control).  If a zero-dose (0 um) dexamethasone control group was utilized in the current experimental design, the authors need to be more explicit in detailing this effort.
  3. Page 3; Paragraph 1; Line 2: Delete “β-Hydroxy-β-Methyl Butyrate” and refer to “HMB”.
  4. Page 3; Paragraph 2; Line 4: Add “an” following “for” and change “additionally” to “additional”.
  5. Page 3; Paragraph 3; Line 1: Change “confluence” to “confluent”.
  6. Page 3; Paragraph 4; Line 2: Change “concentration” to “concentrations”.
  7. Page 3; Paragraph 5; Lines 3 & 5: Change “confluence” to “confluent”.
  8. Page 4; Paragraph 2; Line 2: Change “confluence” to “confluent”.
  9. Page 4; Paragraph 3; Line 1: Change “confluence” to “confluent”.
  10. Page 4; Paragraph 3; Line 7: Change “the block” to “blocking”.
  11. Page 5; Paragraph 2; Line 1: Change “confluence” to “confluent”.

RESULTS

  1. Page5; Paragraph 4; Line 7: Change “...taken...” to “...showed a...”
  2. Page 5; Paragraph 4; Line 8: Add “and” following “1 mM”.
  3. Page 5; Paragraph 4; Lines 9-10: Change “...allowed to observe a synergistic action since the mixture 30uM LA/300 uM HMB was already effective...” to “...allowed for observation of a synergistic dose; and, even though 30 uM LA/300 uM HMB showed effectiveness, it was the 100 uM R(+)LA/1 mM HMB dose that fully restored cell...”

DISCUSSION

  1. Page 14; Paragraph 1; Line 2: Change “optimal” to “most effective” or to “most efficacious”.
  2. Page 14-15; Paragraph 3; Lines 1-3: Change “...were able to concentration-dependently protects muscular cells by dexamethasone-dependent damage, nevertheless the combination allowed synergistic effects about several parameter.” to “...were able to, in a concentration-dependent manner, protect C2C12 muscle cells from dexamethasone-induced damage; and, moreover the combination promoted a synergistic effect regarding several parameters.”
  3. Page 15; Paragraph 1; Line 2: Add “the” prior to “control”.
  4. Page 15; Paragraph 1; Line 7: Change “unbalance” to “imbalance by”.
  5. Page 15; Paragraph 1; Line 9: Change “inducing” to “leading to”.
  6. Page 15; Paragraph 1; Line 14: Change “dangerous” to “deleterious”.
  7. Page 15; Paragraph 1; Line 17: Change “association” to “combination”.
  8. Page 15; Paragraph 1; Line 18: Delete “to” prior to “restore” and Change “of” to “in” following “alterations”.
  9. Page 15; Paragraph 2; Line 17: Delete “as well” following “preserved”.
  10. Page 15; Paragraph 2; Lines 18-20: This last sentence needs to be rewritten for clarity; it is very confusing as it is originally stated.
  11. Page 15; Paragraph 3; Line 1: Change “As regards” to “Regarding”.
  12. Page 15; Paragraph 3; Line 5: Add “mechanism of action” following “pharmacodynamic”.
  13. Page 15; Paragraph 3; Line 6: Write-out “Branch-chain amino acids” prior to using its abbreviation/acronym.
  14. Page 15; Paragraph 3; Lines 7-8: Change “...several studies report that HMB supplementation plays a role...” to “...several studies recently reported that HMB supplementation may present with a novel role...”.
  15. Page 15; Paragraph 3; Line 9: Add “Putative” prior to “HMB”.
  16. Page 16; Paragraph 1; Lines 1-3: Change “...profile, involving anti-oxidant properties and myotube maturation capabilities without intervene on MuRF1 and FoxO signals derangement, is the base for the protective activity...” to “...profile that involves antioxidant properties as well as enhanced myotube maturation suggests to be the basis for the protective effects...
  17. Page 16; Paragraph 2; Line 2: Change “candidate” to “promote”.

REFERENCES

  1. Note above citations included above from reviewer.

TABLE/FIGURE CAPTIONS

None

TABLES/FIGURES

None

Reviewer 3 Report

This manuscript describes some effects of dexamethasone in combination with β-hydroxy-β-methyl butyrate and lipoic acid in a murine skeletal cell line. There are numerous measurements that are appropriate and that yielded quantitative results. Most of the text, especially the Discussion, is fine.

I have the following comments:

  1. Much is known about effects of glucocorticoids on muscle and also about the functional antagonism by β-hydroxy-β-methyl butyrate or α-lipoic acid. This study of in vitro effects of the three compounds show, in a quantitative manner, some of the downstream consequences. The novelty of this report is the combination of β-hydroxy-β-methyl butyrate and lipoic acid in antagonizing the effects of the potent glucocorticoid dexamethasone.
  2. Much is also known about the molecular mechanisms of glucocorticoids and the two agents, β-hydroxy-β-methyl butyrate and lipoic acid. This is described in the Discussion. However, with the exception of caspase-3 and MuRF1 and FoxO, the results in manuscript are rather descriptive; they show merely the downstream consequences. One would, for example, like to know more on the mechanism of β-hydroxy-β-methyl butyrate. Is it metabolized, does it interact with a protein to produce it’s effects? This does not necessarly mean that more experiments have to be performed but a mechanistic insight usually leads to more of them and the results that follow substanciates the mechanistic hypothesis. In the paper “Leucine and HMB Differentially Modulate Proteasome System in Skeletal Muscle Under Different Sarcopenic Conditions (2013)” (1371/journal.pone.0076752 ), that is not cited, an observation is made (“leucine supplementation was unable to provide protection against soleus mass loss in dexamethasone treated rats. Interestingly, HMB supplementation was unable to provide protection against mass loss in all treatments.”) and this is discussed.
  3. Further, it needs to be discussed why lipoic acid has no effect on caspase-3 up to 3mM, whereas β-hydroxy-β-methyl butyrate has a full effect even at 1 mM (Fig 2).
  4. The manuscript is much too long:
    In Materials and methods, most paragraphs start with a description of the cell culture. This must be shortened.
    The same is the case with the figure legends.
    The first sentences of 3.2 repeat detailed experimental conditions in Results.
    Tables 1 and 3 can be fused into one table.
    Table 2 can become Fig. 1A as it shows the same variables as the current Figure 1 that then will be Fig. 1B.
    Further, the cytological staining shown in Figure 4 are of little use. The quantitative assessment in Figure 4F is sufficient.
    The y-axis in Figure 3 does not start at 0. For reasons of comparability, this needs to be corrected.
  5. There are numerous typos: e, g. in the Abstract “In the early phase of myotube the mixture preserved the number of myogenin-positive cells, whereas completely prevent the dexamethasone-dependent damage…” should probably read “In the early phase of myotube formation, the mixture preserved the number of myogenin-positive cells, whereas it completely prevented the dexamethasone-dependent damage…”. The meaning of the sentence “an oxidative modification due to the introduction of carbonyl groups into protein side chains by a site- specific mechanism [42], inducing decreased functionality.” needs improvement. Say what is the chemical reaction and the end-products.
    I cannot comment on all typos but in the uploaded file, where are highlighted some typos and misnomers, some of them can be found.
  6. Some of the literature references are not appropriately relating to the text, e.g. 21, 22 & 62; see the uploaded file. Please check all references for accuracy.
  7. There are too many abbreviations: BSA, PDH, KDH, etc. It is not necessary to introduce an abbreviation when the term is used only 2 or 3 times, see e.g. BSA. Especially in the Abstract, abbreviations should be few if any. Some abbreviations are introduced but not used, e.g. differentiation medium in the Fig legends. On the other hand, it would be good to abbreviate dexamethasone.

Round 2

Reviewer 1 Report

In the revised version of the manuscript, some of the concerns were addressed by authors. However, authors did not address major concerns and other concerns that require additional experiments. Specifically, to change the nature of this manuscript from descriptive to mechanism-oriented, the experiments to propose and prove the molecular mechanism of protective effect of the studied compounds – alone and in combination – should be performed. Otherwise, the level of novelty and scientific significance of this paper will remain low.

As for the other concerns, only concern #4 was addressed properly. Resolution of concerns #3 requires additional experiments to assess apoptosis. Resolution for concern #5 requires completely different set of assessment – confocal microscopy. Concerns #6 and #7 are partially addressed and should be revisited when other concerns are resolved.

Reviewer 2 Report

The authors should be commended on the modifications made to the revised, current manuscript. It now suffices all of this reviewer's comments/concerns. 

One very minor editorial comment; on Page 5, Line 200 of the current pdf revision, "analizying" should be changed to "analyzing"

Reviewer 3 Report

The authors have taken great care to improve the manuscript, both in content as well as in style. Most of the comments I made have been followed to my satisfaction. Some highlights regarding style in the pdf file were perhaps not understood; therefore, I detail these below.

Comments:

Parts of the text are still too wordy; in what follows, I suggest alterations:

Abstract: The last sentence could read: "R(+)LA together with HMB might have potential for the therapy of sarcopenia."

Line 62: the is lacking

Lines 71&72: “The lysine salt of R(+)LA is quite stable and therefore useful in therapeutic applications.”

Line 86: confluency

Lines 105&193 …till…

Line 141: “After extraction, 20 μg samples of protein were denatured in 6% sodium dodecyl sulfate (SDS) and ……”

Line 195: ….by….

Lines >217: "At concentrations of 100 μM LA and 1 mM HMB, full restoration of cellular viability was observed (Figure 1), suggesting synergism.”

Line 247: “DEX at concentrations of 10 and 100 μM appear not to upregulate caspase-3 activity. Please mention and interpret.”

Lines 291&292: this sentence is fine – I couldn’t undo the red highlight.

Lines > 358: “The myotoxicity of DEX depends on cell type and is concentration-dependent with values between 10 nM and 100 μM (REFs). Our result of a threshold value of 1 μM agrees with other reports regarding viability, caspase-3 activity and MuRF1 and FoxO expression (REFs), both of which play ... (54).”

Lines > 382: “In agreement with results from others (13,58), we found that DEX decreased the expression of myogenin in C2C12 myotubes.”

Lines > 390: The statement: “Oxidative stress is a powerful pathological mediator…” is not substantiated by the reference and should be deleted. The text could then be: “Glucocorticoids lead to mitochondrial dysfunction and this has been linked to excessive ROS production (59,60).”

Lines 392-425 and 430-434: The text should be straightened by a person with very knowledge of English.

Lines 426-428: One could say, e.g.: “In summary, the beneficial effects of LA and HMB in counteracting DEX-induced myotoxicity are due to their antioxidant property, their activation of the mTOR cascade and the inhibition of the proteasome pathway.”

Line 433: …. rationale …

Commas are missing in lines: 35,44,49,51,57,59,75,89,236,365

Figures:
Table 3 shows no changes of caspase activity; it should be removed. The text above the table says it all.

In suppl. Table 1, two values are below the control value. Please mention this in Results and interpret.

In the legends to Fig. 3 and suppl. Fig. 3, superoxide anion radical concentrations are given in μM/mg protein; should that be μmol/mg protein?

In the legend to Fig. 3, the phrase "C2C12 myoblasts…." is repeated; please remove and reorder the text.

All 3 suppl. Tables contain minor changes – please correct.
